# Discovery of a small molecule that inhibits bacterial ribosome biogenesis

**Jonathan M Stokes[1], Joseph H Davis[2,3,4], Chand S Mangat[1], James R Williamson[2,3,4], Eric D Brown[1]***

[1]Michael G DeGroote Institute for Infectious Disease Research, Department of Biochemistry and Biomedical Sciences, McMaster University, Hamilton, Canada; [2]Department of Integrative Structural and Computational Biology, The Scripps Research Institute, La Jolla, United States; [3]Department of Chemistry, The Scripps Research Institute, La Jolla, United States; [4]The Skaggs Institute for Chemical Biology, The Scripps Research Institute, La Jolla, United States

**Abstract** While small molecule inhibitors of the bacterial ribosome have been instrumental in understanding protein translation, no such probes exist to study ribosome biogenesis. We screened a diverse chemical collection that included previously approved drugs for compounds that induced cold sensitive growth inhibition in the model bacterium *Escherichia coli*. Among the most cold sensitive was lamotrigine, an anticonvulsant drug. Lamotrigine treatment resulted in the rapid accumulation of immature 30S and 50S ribosomal subunits at 15°C. Importantly, this was not the result of translation inhibition, as lamotrigine was incapable of perturbing protein synthesis in vivo or in vitro. Spontaneous suppressor mutations blocking lamotrigine activity mapped solely to the poorly characterized domain II of translation initiation factor IF2 and prevented the binding of lamotrigine to IF2 in vitro. This work establishes lamotrigine as a widely available chemical probe of bacterial ribosome biogenesis and suggests a role for *E. coli* IF2 in ribosome assembly.

***For correspondence:** ebrown@mcmaster.ca

**Competing interests:** The authors declare that no competing interests exist.

## Introduction

The bacterial ribosome is a 2.6-MDa ribonucleoprotein complex responsible for protein translation, which sediments as a 70S particle composed of a small (30S) and a large (50S) subunit. While there is a relatively thorough understanding of the structure and function of the ribosome during translation (*Moore, 2012*), the molecular events underlying its assembly remain largely enigmatic. Ribosome biogenesis, which consumes up to 40% of the cell's energy in rapidly growing *Escherichia coli* (*Maguire, 2009*), involves the coordinated transcription, modification, and folding of rRNA transcripts; translation, modification, and folding of r-proteins; binding of r-proteins to the appropriate rRNA scaffolds; and binding and release of ribosome biogenesis factors. In vivo, these events occur in parallel and represent a highly dynamic system of interrelated processes that occur cooperatively to narrow the assembly landscape of the ribosome (*Holmes and Culver, 2005*; *Williamson, 2005*; *Kim et al., 2014*).

Ribosome biogenesis factors are proteins that transiently bind to assembling ribosomal particles to increase the efficiency of subunit maturation (*Bunner et al., 2010*) and prevent the entry of immature subunits into the translation cycle (*Strunk et al., 2011*; *Boehringer et al., 2012*; *Lebaron et al., 2012*; *Strunk et al., 2012*). *E. coli* has approximately 60 of such factors. Genetic perturbation has been the conventional route to probe the function of these proteins but has drawbacks. Genetic inactivation is typically permanent, often 'all or none' in scope, and for essential genes is fraught with the difficulty of creating conditional alleles. Further, due to the coordination of 30S and 50S subunit biogenesis, and regulatory feedback from the translational capacity of the cell (*Yamagishi and Nomura, 1988*; *Gaal et al., 1997*),

**eLife digest** Inside cells, molecular machines called ribosomes make proteins from instructions that are provided by genes. The ribosomes themselves are made up of about 50 proteins and three RNA molecules that need to be assembled like a 3-D jigsaw. In bacteria, a group of proteins called ribosome biogenesis factors help to assemble these pieces correctly.

To study how a biological process works, scientists often look at what happens when a component is missing or not working properly. However, this approach cannot be used to study how ribosomes are made because stopping protein production entirely will kill the cell. Another approach is to use chemicals to temporarily stop or slow down a biological process, but researchers are yet to find a chemical that can do this for ribosome assembly.

To address this problem, Stokes et al. 'screened' 30,000 chemicals in an effort to find one or more that could affect ribosome assembly in bacteria. The screen revealed that a drug called lamotrigine—which is used to treat epilepsy and other conditions in humans—could stop the assembly of ribosomes, but did not affect the production of proteins by completed ribosomes.

The experiments also suggest that initiation factor 2, a protein that is involved in the production of other proteins, may also have a role in ribosome assembly. Another recent study found that the equivalent of initiation factor 2 in yeast acts as a quality control checkpoint during ribosome assembly, so the bacterial version may also perform a similar role.

It is also be possible that lamotrigine might be used to help develop a novel mechanistic class of antibiotics.

genetic probes of ribosome assembly are prone to wide-ranging impacts and pleiotropic phenotypes (*Lerner and Inouye, 1991*).

Small molecules are finding increasing use in a research paradigm that emphasizes the value of these as probes of biology. Such chemicals can exert their effects on a time scale of seconds and be added or removed from cell systems at will. Further, small molecules can be dosed to achieve varying levels of target inhibition and as such can be elegant probes of protein function. While existing antibiotics provide a surfeit of probes for on-going efforts to understand the mechanistic details of protein translation, no chemical probes exist for the study of ribosome biogenesis. Small molecule inhibitors of ribosome biogenesis could provide important new tools for the study of this complex process, particularly those events controlled by uncharacterized protein assembly factors. Additionally, chemical inhibitors of bacterial ribosome biogenesis might serve as leads for an entirely new mechanistic class of antibiotics (*Comartin and Brown, 2006*).

In this study, we report the discovery and characterization of a chemical inhibitor of bacterial ribosome biogenesis. Using a diverse chemical library that included previously approved drugs and compounds of known bioactivity, we enriched for molecules that induced cold sensitive growth inhibition in the model bacterium *E. coli*. Indeed, numerous studies have revealed that genetic defects in ribosome assembly result in cold sensitive growth phenotypes (*Bryant and Sypherd, 1974*; *Dammel and Noller, 1995*; *Jones et al., 1996*; *Bubunenko et al., 2006*; *Connolly et al., 2008*; *Clatterbuck Soper et al., 2013*). We too performed validating efforts, reported herein, of the cold sensitivity of strains from the Keio collection, a comprehensive compendium of *E. coli* deletion strains. A subsequent chemical screen determined that the anticonvulsant drug lamotrigine induced a strongly cold sensitive growth phenotype. Treatment with this molecule resulted in the accumulation of immature ribosomal subunits in a time-dependent manner without inhibiting protein translation. Spontaneous suppressors of lamotrigine activity mapped exclusively to translation initiation factor IF2, encoded by *infB*. These mutations, found in the poorly characterized and evolutionarily divergent domain II of IF2, obviated the binding of lamotrigine to IF2 in vitro. This work establishes lamotrigine as a widely available chemical probe of bacterial ribosome biogenesis and suggests a role for *E. coli* IF2 in this process.

## Results

### The ribosome is a primary target of cold stress

Where cold sensitive growth has previously been identified as a dominant phenotype for defects in ribosome biogenesis, we set out to first validate such an enrichment strategy with a screen of the *E. coli* Keio

collection (*Baba et al., 2006*), a comprehensive set of non-essential gene deletion strains (*Figure 1—source data 1*). We looked for strains that were sensitized to growth at 15°C compared to 37°C (*Figure 1—figure supplement 1A,B*). A cold sensitivity factor was subsequently generated for each clone, defined as the ratio of growth at 37°C to growth at 15°C, normalized to the mean growth ratio measured for the entire collection (*Figure 1A*). Strains that displayed a cold sensitivity factor in the top 3.5% (155 clones) were analyzed using clusters of orthologous groups (*Tatusov et al., 1997*, *2003*) to categorize the cellular function of each deleted gene (*Figure 1—figure supplement 1C*, *Supplementary file 1A*). To highlight the relative proportion of genes in each functional class, the number of cold sensitive genes in each was divided by the total number of non-essential genes in that same category (*Figure 1B*). This normalization procedure highlighted ribosome-related genes as exceptionally sensitive

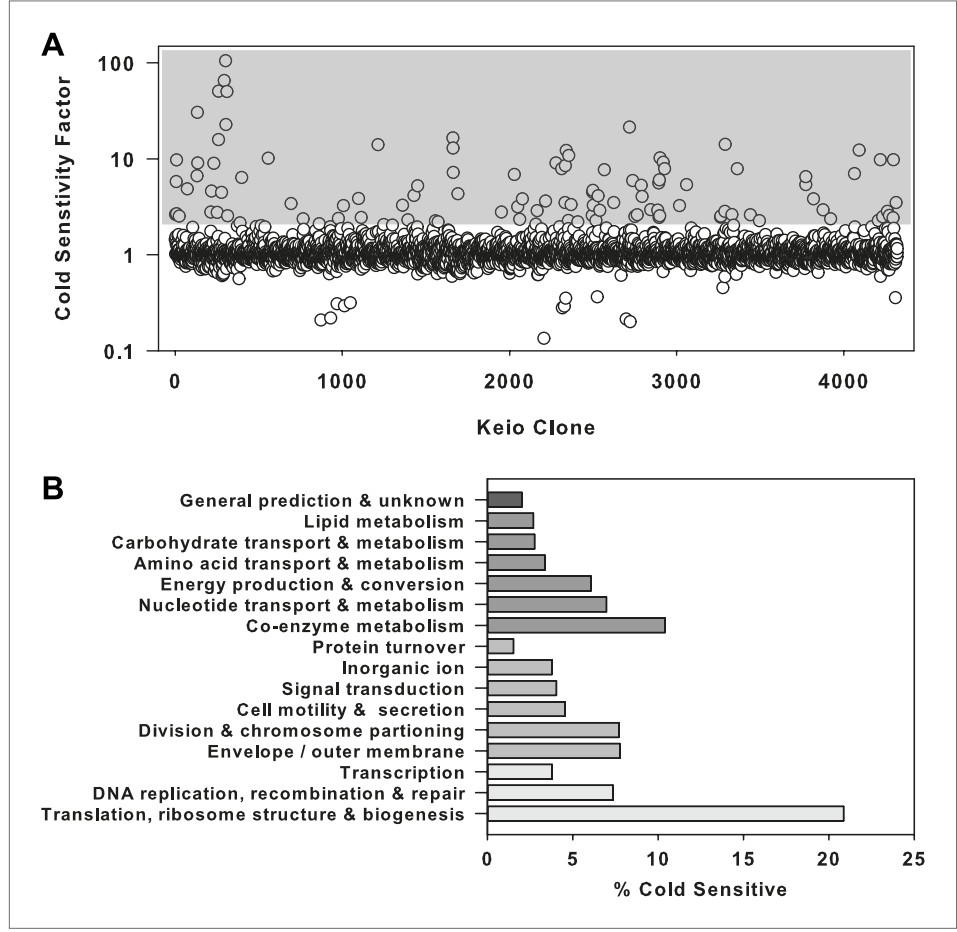

**Figure 1**. The ribosome is a primary target of cold stress. (**A**) Screen of the *E. coli* Keio collection for cold sensitivity. Each strain's cold sensitivity factor is defined as the ratio of growth at 37°C to growth at 15°C. Cold sensitivity factors for each strain were normalized to 1, based on the mean of all cold sensitivity factors calculated for the entire collection. Growth at each temperature was calculated based on the average of two replicates. A gray box highlights strains exhibiting cold sensitivity in the top 3.5% (155 strains). (**B**) The 155 cold sensitive genes from (**A**) were grouped based on clusters of orthologous groups classifications. The percentage of cold sensitive genes in each functional class was defined as the number of cold sensitive genes in that class divided by the total number of non-essential *E. coli* genes in that same functional class. By permuting the classification assignments, we determined that the proportion of cold sensitive genes in the translation class (21%) was significant with a bootstrapped p-value < 1e−6.

The following source data and figure supplement are available for figure 1:

**Source data 1**. Screen of the *E. coli* Keio collection.

**Figure supplement 1**. Primary data from the screen of the *E. coli* Keio collection.

to low temperatures, as >20% of genes in this functional class were found to be cold sensitive. Importantly, this screen was also successful in identifying the vast majority of previously reported cold sensitive ribosome biogenesis genes (*Supplementary file 1B*), providing support that screening compounds for cold sensitivity would enrich for those related to ribosome function and biogenesis.

## Lamotrigine induces profound cold sensitivity in *E. coli*

Having validated our cold sensitivity enrichment strategy, we proceeded to screen a diverse chemical collection to identify molecules that exhibited a cold sensitive growth inhibition phenotype. This collection, assembled from a variety of vendors, included some 30,000 compounds. These were largely diverse synthetic molecules with a subset of 3500 previously approved drugs and chemicals with known biological activity (*Figure 2—source data 1*). *E. coli* was grown in LB media at 15°C and 37°C in the presence of 10 μM of each compound (*Figure 2—figure supplement 1*). To select compounds for follow-up, we identified those that strongly inhibited growth at 15°C (>3σ below the mean $OD_{600}$ at 15°C), yet displayed little growth inhibition at 37°C (<2σ below the mean $OD_{600}$ at 37°C). These criteria resulted in 49 active compounds (*Figure 2A*). We removed all antibiotics with known mechanisms of action and filtered the active molecules for diversity in chemical structure. This led to a short-list of 38 active molecules, which were analyzed in dose at 37°C and 15°C. The anticonvulsant drug lamotrigine displayed the largest change in minimum inhibitory concentration (MIC) upon temperature downshift, increasing in potency more than 50-fold from >512 μM at 37°C to 7.8 μM at 15°C (*Figure 2B*, *Figure 2—figure supplement 2*).

## Lamotrigine treatment results in the accumulation of non-native ribosomal particles

To determine whether lamotrigine resulted in cold sensitivity through perturbation of the ribosome, we harvested ribosomal particles from early-log cultures of *E. coli* treated with 2× MIC of lamotrigine for 1 hr and 6 hr at 15°C in LB media and resolved them using sucrose density centrifugation. We note that the doubling time of wild-type *E. coli* at 15°C in LB media was 6 hr. Cultures were pulse labeled with [14C]-uridine immediately upon drug treatment to visualize the accumulation of newly synthesized particles. Since previous reports have shown that inhibitors of protein translation can cause accumulation of immature ribosomal particles (*Siibak et al., 2009*, *2011*; *Sykes et al., 2010*), we also tested a panel of antibiotics (*Figure 3—figure supplement 1A–D*) with known mechanism of action for comparison.

Mock treatment of cells with DMSO and simultaneous pulsing with [14C]-uridine allowed for the visualization of 30S, 50S, and 70S particle accumulation after 1 hr and 6 hr of growth post-treatment (*Figure 3A*). After 1 hr of treatment, small quantities of newly synthesized particles were present, and after 6 hr, cells had accumulated labeled ribosomal particles to near steady-state levels. Cultures treated with chloramphenicol (*Figure 3B*), erythromycin (*Figure 3C*), and tetracycline (*Figure 3D*) displayed a substantial accumulation of non-native ribosomal particles after just 1 hr of treatment, illustrating that inhibition of translation can indirectly inhibit ribosomal subunit assembly by limiting the availability of r-proteins. Interestingly, we found that the addition of 2× MIC of vancomycin to *E. coli* resulted in a detectable perturbation of the ribosome profile (*Figure 3E*). However, the presence of a ~40S particle after 6 hr of treatment is likely the result of cell lysis induced by the inhibition of peptidoglycan synthesis (Stokes and Brown, unpublished data). Treatment with lamotrigine resulted in the accumulation of non-native ribosomal particles after 1 hr of incubation and did so in a time-dependent manner (*Figure 3F*). Further investigations revealed that treatment of *E. coli* with 2× MIC of lamotrigine for only 5 min (~1% of the doubling time) caused a significant accumulation of these non-native particles (*Figure 3G,H*). Consistent with the cold sensitive phenotype induced by lamotrigine, these pre-30S and pre-50S particles that accumulated at 15°C were not present after treatment at 37°C (*Figure 3—figure supplement 1E,F*).

To determine if lamotrigine directly associated with ribosomal particles, early-log cultures of *E. coli* were treated with [3H]-lamotrigine in the absence (*Figure 3—figure supplement 1G*) or presence (*Figure 3—figure supplement 1H*) of 1× MIC of unlabeled lamotrigine. Ribosomal particles were then separated on a sucrose gradient and individual fractions counted to localize [3H]-lamotrigine. Radiolabeled compound was found exclusively in the soluble fractions eluting early in the gradient, suggesting that lamotrigine does not interact directly with mature or non-native ribosomal particles.

## Non-native ribosomal particles are immature 30S and 50S subunits

We reasoned that ribosomal particles accumulating during treatment could be immature subunits or degradation products of weakly assembled ribosomes. Thus, we analyzed rRNA processing and r-protein

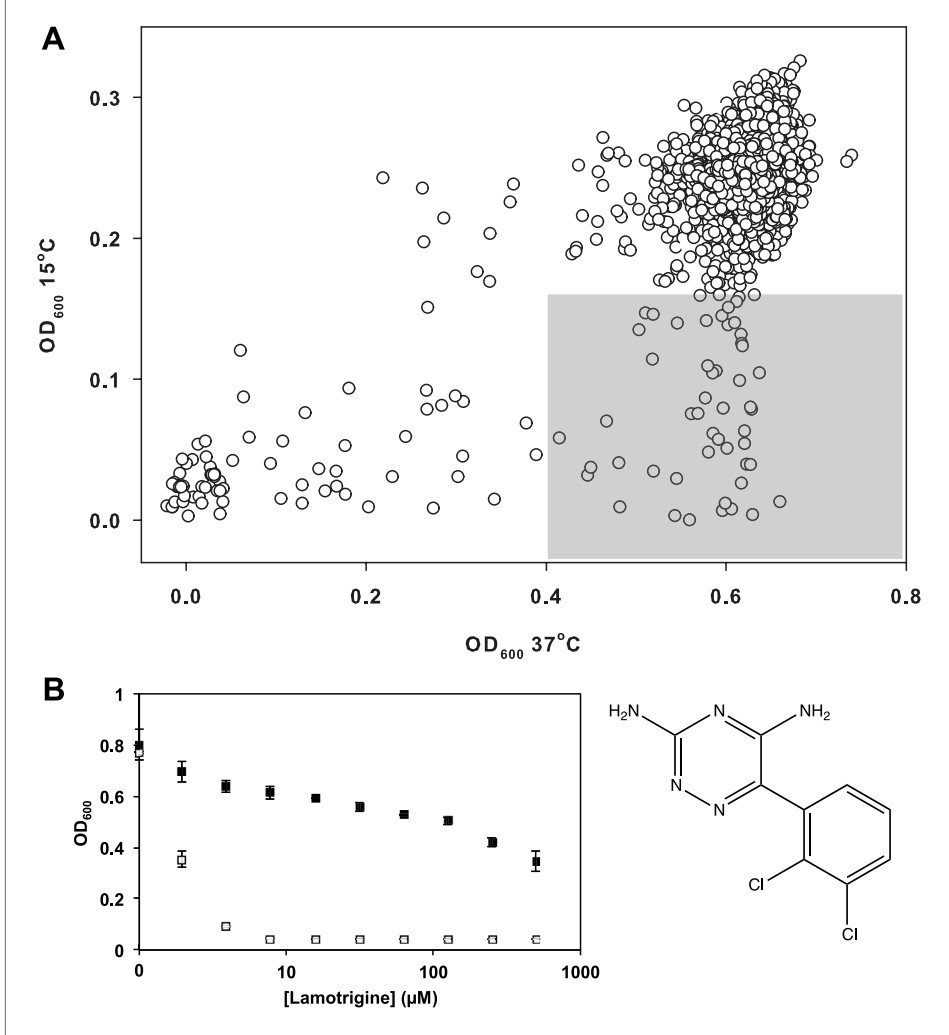

**Figure 2**. Lamotrigine induces profound cold sensitivity in *E. coli*. (**A**) Screen of ~30,000 small molecules at 10 μM against *E. coli* for cold sensitivity. Compounds found within the gray region were selected for secondary screening. Hit inclusion boundaries are defined as molecules residing >3σ below the mean $OD_{600}$ at 15°C and <2σ below the mean $OD_{600}$ at 37°C. Growth at each temperature was calculated based on the average of two replicates. (**B**) Dose-response analysis of lamotrigine at 37°C (black dots) and 15°C (white dots). Error bars represent the error of two biological replicates.

The following source data and figure supplements are available for figure 2:

**Source data 1**. Small molecule screen for cold senstivity.

**Figure supplement 1**. Primary data from the small molecule screen.

**Figure supplement 2**. Temperature dependence of lamotrigine activity in *E. coli*.

---

content of all particles that accumulated upon lamotrigine treatment. Because previous investigations have shown that the cleavage of 5′ and 3′ termini of rRNA is among the final events in ribosomal subunit assembly (***Lindahl, 1973***; ***Mangiarotti et al., 1974***; ***Srivastava and Schlessinger, 1988***), we first performed 5′ primer extension reactions using rRNA purified from sucrose gradients of lamotrigine-treated cells. Early-log cultures of *E. coli* were grown in the presence of 2× MIC of lamotrigine at 15°C in LB media for 5 min, 1 hr, and 6 hr, at which time the ribosomal particles were resolved on sucrose gradients, and the rRNA corresponding to each discrete particle was purified and reverse transcribed using 5′ carboxyfluorescein-tagged primers. A 16S rRNA-specific primer was used to analyze

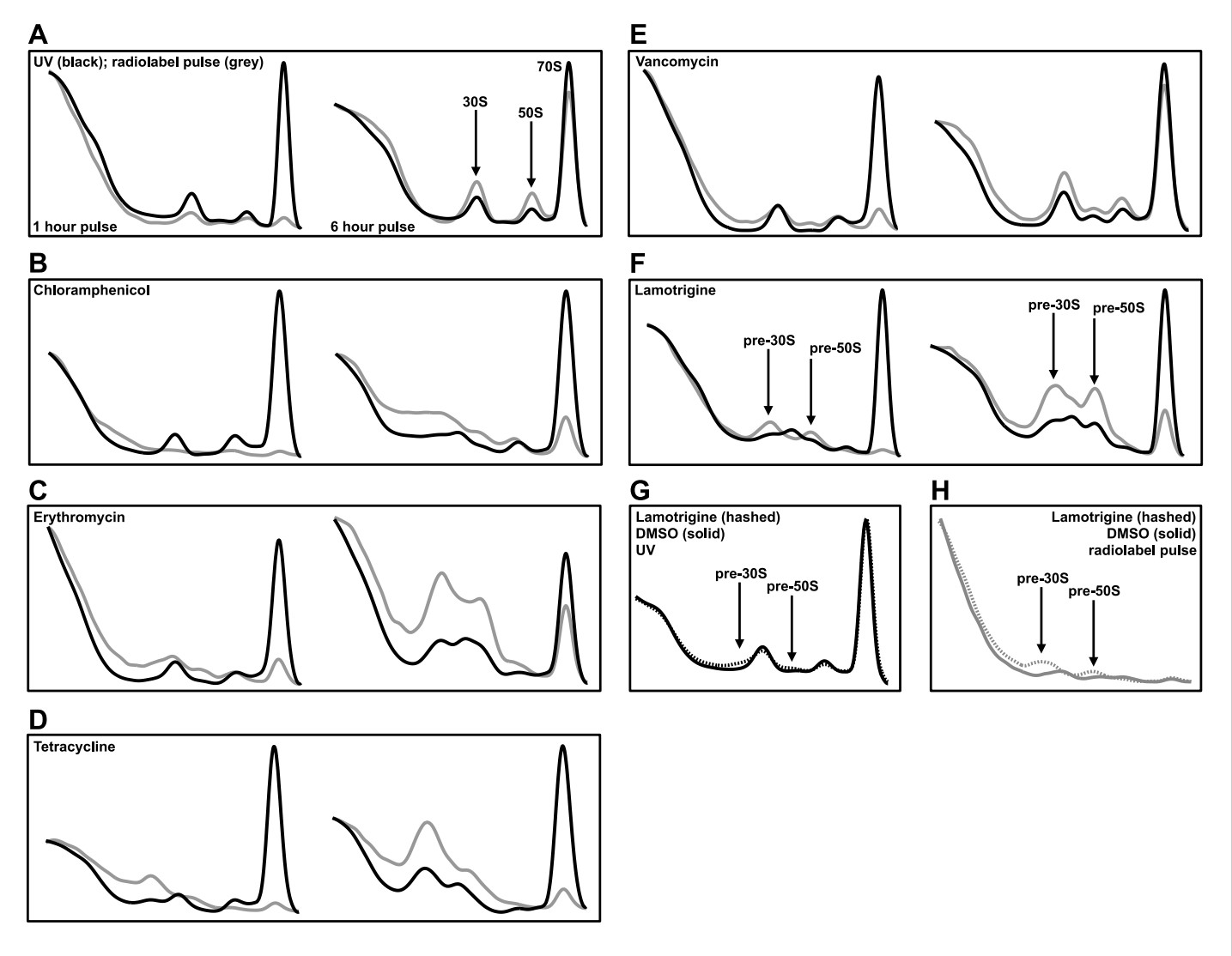

**Figure 3**. Lamotrigine treatment results in the accumulation of non-native ribosomal particles. (**A**) Cells were treated with DMSO (vehicle) and immediately pulse labeled with [$^{14}$C]-uridine. Cells were harvested after 1 hr (left) and 6 hr (right) of treatment, and ribosomal particle accumulation was monitored using UV absorbance at 260 nm (black trace) and scintillation counting (gray trace). Also shown are treatments with 2× MIC chloramphenicol (**B**); 2× MIC erythromycin (**C**); 2× MIC tetracycline (**D**); 2× MIC vancomycin (**E**); and 2× MIC lamotrigine (**F**). (**G**) Early-log cultures of *E. coli* were treated with DMSO (solid line) or 2× MIC lamotrigine (hashed line), pulse labeled with [$^{14}$C]-uridine, and incubated for 5 min. Ribosomal particles were separated on a sucrose gradient and monitored using UV absorbance. (**H**) These gradients were also analyzed via scintillation counting.
The following figure supplement is available for figure 3:

**Figure supplement 1**. Temperature-dependent antibiotic activity in *E. coli*.

the 30S subunit rRNA in pre-30S, 30S, and 70S fractions, whereas a 23S rRNA-specific primer was used to analyze the 50S subunit rRNA in pre-30S, pre-50S, 50S, and 70S fractions. **Figure 4A** displays the 5′ cleavage events during the processing of 16S and 23S rRNAs (**Shajani et al., 2011**). We note here that our experiments were unable to detect the first 16S cleavage event of 49 nucleotides by Rnase E, and that all immature 16S rRNA species described contain a full-length 5′ terminus of 115 nucleotides.

Primer extension analysis of cells treated with DMSO for 6 hr is depicted in **Figure 4B** (top panel). Analysis of the pre-30S, 30S, and 70S regions of the gradient using the 16S rRNA-specific primer revealed that increasing sedimentation rate correlated with a decreased proportion of immature 16S rRNA relative to total 16S rRNA. The presence of immature 16S rRNA sedimenting in the pre-30S

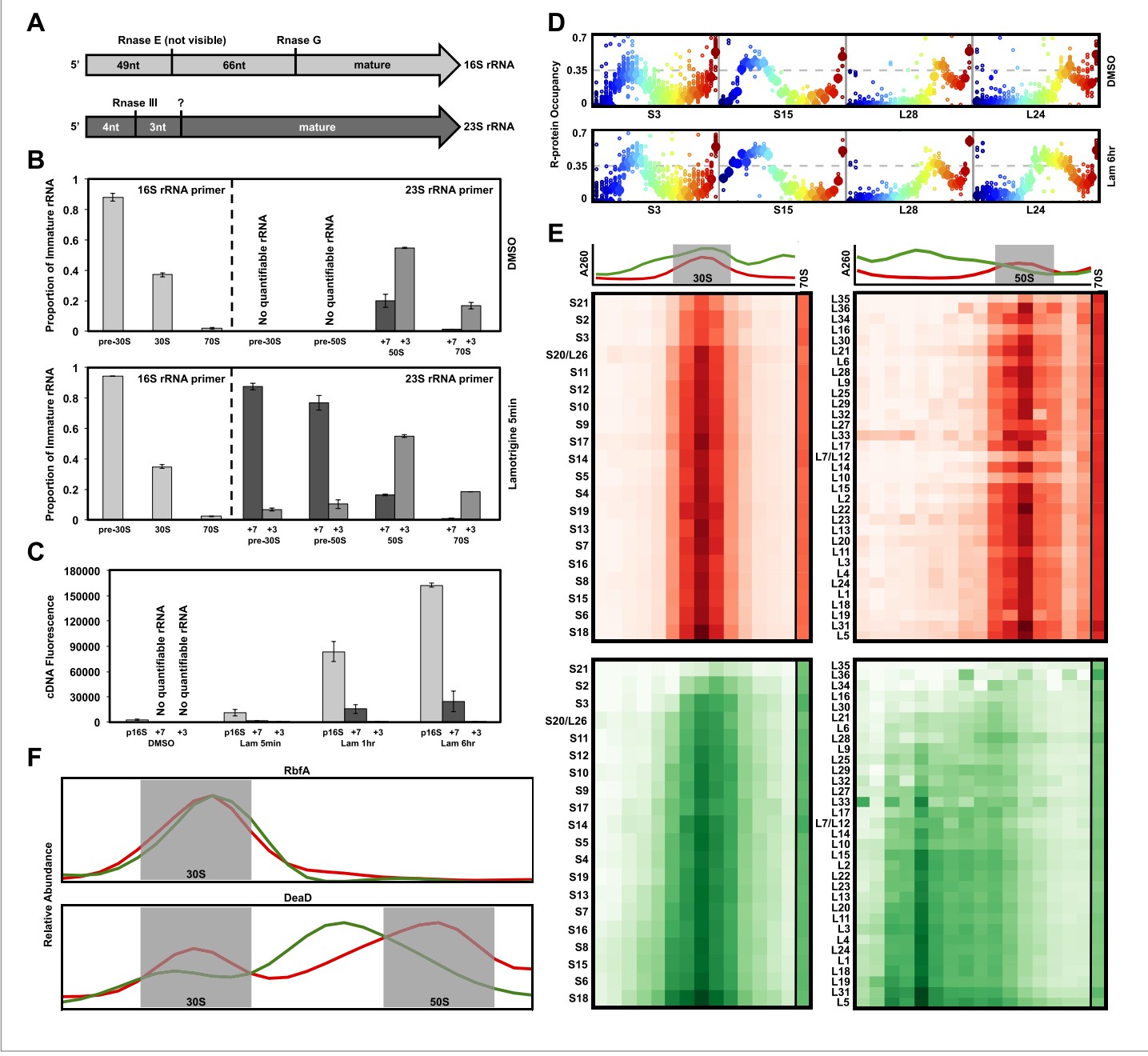

**Figure 4**. Non-native ribosomal particles are immature 30S and 50S subunits. (**A**) 5′ cleavage sites of 16S and 23S rRNA. (**B**) 5′ primer extension analysis of ribosomal particles harvested from DMSO- and lamotrigine-treated *E. coli*. Early-log cells were treated with DMSO for 6 hr or 2× MIC lamotrigine for 5 min, ribosomal particles were separated on a sucrose gradient, and rRNA was fractionated according to increasing sedimentation rates as indicated (pre-30S, 30S, pre-50S, 50S, and 70S). Particle detection by reverse transcription used a 16S rRNA specific primer (light gray) or a 23S rRNA specific primer (gray and dark gray). Proportion of immature rRNA was calculated as (immature 16S rRNA species/total 16S rRNA species) and (immature 23S rRNA species/total 23S rRNA species). +7 and +3 represent immature 23S rRNA containing an additional 7 nucleotides and 3 nucleotides at the 5′ terminus, respectively. Error bars represent the error of two biological replicates. (**C**) Quantitative cDNA production of rRNA species within pre-30S regions from DMSO- and lamotrigine-treated *E. coli*. Early-log cells were treated with DMSO for 6 hr or 2× MIC lamotrigine for 5 min, 1 hr, and 6 hr. Ribosomal particles were separated on a sucrose gradient, and rRNA purified from a single pre-30S fraction from each treatment was reverse transcribed in parallel using 16S- and 23S-specific primers. p16S represents immature 16S rRNA. Error bars represent the error of two biological replicates. (**D**) Quantitation of ribosomal protein occupancy within individual fractions collected from sucrose gradients. Fractions are colored from blue (lowest density portion of the gradient) to red (highest density portion of the gradient). Each open circle represents a unique peptide measurement; closed circles denote median values. Occupancy profiles for early (S15, L24) and late binding (S3, L28) proteins are compared between sucrose gradients analyzed using DMSO (top)

*Figure 4. Continued on next page*

*Figure 4. Continued*

or lamotrigine-treated (bottom) cells. (**E**) R-protein occupancy of ribosomal particles harvested from sucrose density gradient fractions of DMSO- (red) and lamotrigine-treated (green) *E. coli*. Data are plotted as a heat map using the median occupancy values (see results) corrected for the amount of sample analyzed in each fraction and normalized to scale from 0 (white) to 1.0 (darkest shade). Small subunit (left) fractions span the pre-30S to the pre-50S regions of the sucrose gradient. Large subunit fractions (right) span the late-30S to the late-50S regions of the sucrose gradient. A representative 70S fraction is included in each data set. Absorbance measured at 260 nm is plotted for the region analyzed above each heat map. (**F**) Mass spectrometric localization of RbfA and DeaD in sucrose density gradient fractions of DMSO- (red) and lamotrigine-treated (green) *E. coli*. Relative protein abundance was calculated as described in 'Materials and methods'.

The following source data and figure supplements are available for figure 4:

**Source data 1**. R-protein occupancy across sucrose gradients, normalized to the maximum value observed.
**Figure supplement 1**. 5′ primer extension of lamotrigine-treated *E. coli*.
**Figure supplement 2**. R-protein mass spectrometry of ribosomal particles from lamotrigine-treated *E. coli*.

region suggested a heterogeneous composition of 30S particles at various stages of maturation. At the maximum of the 30S peak approximately 40% of 16S rRNA was unprocessed. The 16S rRNA found in fractions corresponding to the 70S subunit was >95% processed, as expected. Overall, a similar trend was seen when analyzing the processing of 23S rRNA in the pre-30S, pre-50S, 50S, and 70S regions of the gradient; increasing sedimentation rate correlated with a decreased proportion of immature 23S rRNA relative to total 23S rRNA. The pre-30S and pre-50S regions of the gradient were devoid of quantifiable 23S rRNA, suggesting very little 50S precursor accumulation in unperturbed cells.

Compared to cells treated with DMSO, those treated with 2× MIC of lamotrigine for 5 min (*Figure 4B*, bottom panel), 1 hr (*Figure 4—figure supplement 1A*), and 6 hr (*Figure 4—figure supplement 1B*) contained similar proportions of immature to total 16S and 23S rRNA in the 30S, 50S, and 70S regions of the gradient. Furthermore, lamotrigine-treated cells displayed almost identical proportions of immature to total 16S rRNA in the pre-30S region, relative to DMSO-treated cells. Unlike DMSO-treatment, however, lamotrigine treatment resulted in the accumulation of immature 23S rRNA in the pre-30S and pre-50S regions. While the presence of unprocessed 23S rRNA in the pre-50S region strongly suggested an immature 50S subunit sedimenting at ~40S, its presence in the pre-30S region raised questions of whether the dominant species in the pre-30S peak (*Figure 3F*) was derived from 16S or 23S rRNA.

While calculating proportions of immature to total rRNA of the same species ([immature 16S/total 16S] and [immature 23S/total 23S]) provides detail of rRNA processing efficiency in each region of the gradient, it does not inform on the absolute quantity of one species (16S rRNA) relative to the other (23S rRNA). To answer this question, we quantified absolute cDNA fluorescence from 5′ primer extension reactions (*Figure 4C*, *Figure 4—figure supplement 1C*). Samples of rRNA purified from single fractions of sucrose gradients were reverse transcribed in parallel reactions using either the 16S- or 23S-specific primer. In this study, cDNA production is proportional to the amount of rRNA transcript in the sample and therefore reflects absolute quantities of each rRNA species (16S and 23S) present. The quantity of immature 16S rRNA from DMSO-treated cells was minor relative to cells treated with lamotrigine. This is consistent with previous reports, which have shown that immature ribosomal particles in unperturbed cells account for only a small proportion of total ribosomal material (*Mulder et al., 2010*; *Chen et al., 2012*). Furthermore, while immature 23S rRNA slowly accumulated in this region as a function of lamotrigine treatment length, immature 16S rRNA did so at a significantly greater rate. These results indicate that, by far, the major species of rRNA residing within the pre-30S region in lamotrigine-treated cells was unprocessed 16S rRNA. Thus, 5′ primer extension results strongly suggested that lamotrigine treatment results in the accumulation of an immature 30S subunit that sediments at ~25S and an immature 50S subunit that sediments at ~40S.

To further test this hypothesis, we used quantitative mass spectrometry to determine the relative stoichiometry of r-proteins across sucrose gradients of DMSO- and lamotrigine-treated cells. Early-log cultures of *E. coli* grown at 15°C in $^{14}$N-labeled LB media were treated with DMSO or 2× MIC of lamotrigine for 6 hr, lysed, and the ribosomal particles were separated through sucrose gradients. Fractions spanning the pre-30S to the 70S regions were spiked with a fixed concentration of 70S ribosomes purified from cells grown in $^{15}$N-labeled media. These spiked samples were then digested with trypsin

and prepared for mass spectrometry. This approach resulted in multiple independent peptide measurements for each r-protein in every fraction (*Figure 4D*, *Figure 4—figure supplement 2A,B*). Protein occupancy was calculated as $^{14}N/[^{14}N + ^{15}N]$. Direct inspection of the protein occupancy profiles revealed distinct patterns for early- (e.g., S15, L24) and late- (e.g., S3, L28) binding proteins. In the DMSO samples, all r-proteins within a given subunit displayed highly correlated occupancy patterns with maximal occupancy corresponding to 'peak' fractions as determined by rRNA absorbance (*Figure 4D*). In contrast, treatment with lamotrigine resulted in significant occupancy of the early-binding proteins in pre-30S and pre-50S fractions, whereas the late-binding r-proteins exhibited relatively unperturbed profiles. Indeed, protein S15 is found at significantly greater occupancy in the pre-30S fractions upon lamotrigine treatment (dark blue). This effect on early binding proteins is particularly pronounced in the L24 profile with peak occupancy shifted six fractions earlier in the gradient (from orange to green). To facilitate further analysis, this large data set (~20,000 measurements) was compressed to a 53-protein × 28-fraction heat map using the median protein occupancy value for each protein in each fraction (*Figure 4E*, *Figure 4—source data 1*). As expected, the 70S peak from both DMSO- and lamotrigine-treated cells exhibited stoichiometric occupancy of each r-protein.

In both DMSO- and lamotrigine-treated samples, we observed sub-stoichiometric occupancy of the late-binding r-proteins S21, S2, and to a lesser extent S3, within the 30S peak. Notably, this effect was enhanced in the lamotrigine-treated samples. The depletion of these proteins is consistent with prior in vivo analysis of small subunit biogenesis at 37°C, which found S2, S3, and S21 to be the latest-binding small subunit proteins (*Figure 4—figure supplement 2C*; *Chen and Williamson, 2013*). Consistent with sucrose gradient traces monitoring UV absorbance and $[^{14}C]$-uridine incorporation, we observed a subtle broadening of the 30S protein occupancy peak upon treatment with lamotrigine. Analysis of the leading edge of this peak revealed an enrichment of relatively early-binding proteins S13, S7, S16, S8, S15, S6, and S18, consistent with the presence of 30S subunit assembly intermediates in lamotrigine-treated cells. Inspection of large subunit protein occupancy revealed drastic changes as a result of lamotrigine treatment, resulting in the accumulation of an immature particle depleted of the late-binding r-proteins L35, L36, L16, L30, and L28 as well as earlier-binding proteins, L34, L6, and L21 (*Figure 4—figure supplement 2D*). Particles with this heterogeneous protein composition could not be found in the DMSO-treated samples, indicating that if they do form they rapidly convert to mature particles that migrate later in the gradient.

Formally, these r-protein occupancy patterns could have resulted either from immature assembly intermediates or from the degradation of mature particles initiated by the removal of late-binding proteins. To distinguish between these possibilities, we used mass spectrometry to determine the occupancy pattern for the ribosome biogenesis factors RbfA and DeaD (*Figure 4F*). These proteins were completely absent from mature 70S particles, and thus we reasoned that their presence could be used as markers of immature particles. The 30S-specific maturation factor RbfA co-sedimented with the 30S particles in both DMSO- and lamotrigine-treated samples. Further, the 50S-biogenesis factor DeaD co-migrated with the pre-50S peak in the lamotrigine-treated samples and the 50S peak in the DMSO-treated samples. These data further suggest that the pre-30S and pre-50S particles are immature subunits and not the result of degradation of mature ribosomal particles.

## Lamotrigine binds to wild type but not mutant IF2 in a G-nucleotide-dependent manner

To identify lamotrigine's target in vivo, we generated suppressor mutants and sequenced the resulting genomes to identify the mutation(s) that were responsible for resistance. Briefly, *E. coli* BW25113 was grown at 15°C in the presence of 5× MIC (39 µM) of lamotrigine in LB media to saturation. Putative suppressors were then serially passaged in the presence and absence of lamotrigine to purify and to ensure mutation stability. After 20 independent strains had been isolated, three were selected at random, sequenced using an Illumina MiSeq platform, and analyzed against the *E. coli* MG1655 genome using BreSeq. At this time, the chromosome of BW25113 had yet to be sequenced, thus this strain was sequenced in parallel to be used as a reference genome. The sequencing data revealed mutations solely in domain II of initiation factor IF2 (*Figure 5A*, *Figure 5—figure supplement 1A*). Subsequent Sanger sequencing of the *infB* genes from each of the remaining 17 suppressor strains revealed that all mutations mapped to domain II of IF2 and fell into only four categories. Three classes of mutant contained in-frame chromosomal deletions in this region and one mutant class contained a short duplication.

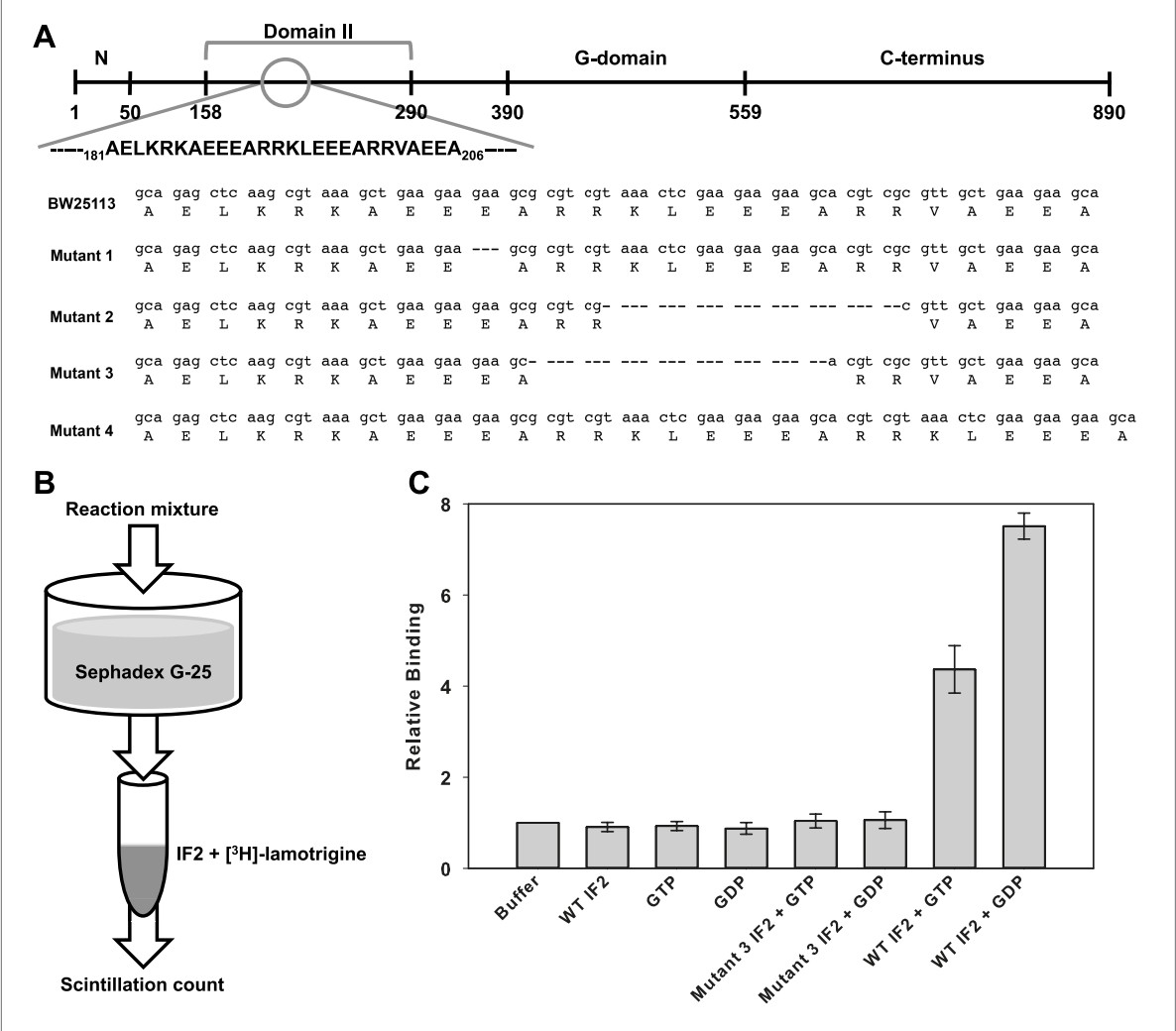

**Figure 5**. Lamotrigine binds to wild type but not mutant IF2 in a G-nucleotide-dependent manner. (**A**) General domain organization of Enterobacteriaceae IF2-α with lamotrigine suppressor mutations mapped against the parental *E. coli* BW25113 sequence. (**B**) Experimental design of [3H]-lamotrigine association assay. (**C**) Relative association of [3H]-lamotrigine to wild-type *E. coli* IF2 and lamotrigine suppressor IF2 (mutant #3) under varying conditions. CPMs of the experimental samples were normalized to the baseline flow-through of [3H]-lamotrigine in buffer. Error bars represent the error of three biological replicates.

The following figure supplement is available for figure 5:

**Figure supplement 1**. Genetic determinants of lamotrigine activity.

To understand the phenotypic characteristics of these four unique lamotrigine suppressor strains, cells from each class were first analyzed for growth rate and resistance to lamotrigine. All displayed wild-type growth at 15°C in the absence of lamotrigine and little sensitivity to lamotrigine treatment up to 512 μM in LB media (***Figure 5—figure supplement 1B,C***). We subsequently analyzed these strains to determine the composition of ribosomal particles upon lamotrigine treatment. Suppressor strains were grown to early-log phase, treated with 2× MIC of lamotrigine, and grown for 6 hr at 15°C. Immediately after the addition of lamotrigine to the cultures, cells were pulse labeled with [14C]-uridine to monitor accumulation of non-native ribosomal particles. Treatment of suppressor strains with lamotrigine did not result in the accumulation of non-native ribosomal particles (***Figure 5—figure supplement 1D***).

To test the hypothesis that IF2 was the target of lamotrigine, we conducted in vitro binding studies using recombinant *E. coli* IF2 and [3H]-lamotrigine (***Figure 5B***). Wild type and mutant forms of *E. coli* IF2 were purified and incubated with [3H]-lamotrigine in the presence of GDP or GTP. After incubation

for 3 hr at 15°C, the reaction mixtures were passed through a pre-cooled Sephadex G-25 column, and the flow-through was collected and scintillation counted to detect the presence of lamotrigine-IF2 complexes. Lamotrigine was found to associate with wild-type *E. coli* IF2 in a G-nucleotide-dependent manner, with lamotrigine-IF2 complex formation favored in the presence of GDP over GTP (*Figure 5C*). We note here that previous studies have not reported measurable GTP turnover by IF2 in the absence of ribosomal subunits (*Severini et al., 1991*). Analyses of lamotrigine binding with mutant IF2 in the presence of GTP and GDP failed to reveal association (*Figure 5C*). Consistent with these results, lamotrigine was found to have activity solely against members of the Enterobacteriaceae (*Supplementary file 2*), which is the only bacterial family that contains the domain II sequence outlined in *Figure 5A*. Interestingly, the potency of lamotrigine at 15°C against the Enterobacteriaceae was directly correlated with IF2-α sequence homology, further suggesting that the sequence of domain II defines essential structural features for lamotrigine association (*Figure 5—figure supplement 1E,F*).

## Accumulation of immature ribosomal subunits is not the result of translation inhibition

Given the known role of IF2 in protein translation, we tested whether lamotrigine was indirectly perturbing ribosome biogenesis by inhibiting IF2-dependent translation. We first monitored [$^{35}$S]-methionine incorporation into bulk cellular protein. Early-log cultures of *E. coli* grown in M9 minimal media were treated with lamotrigine and a collection of known antibiotics for 2.6 hr at 15°C (doubling time = 16 hr). Immediately after the addition of drug, cells were pulsed with [$^{35}$S]-methionine to monitor the production of newly synthesized proteins. Cells were then pelleted, washed, lysed, and treated with TCA. The precipitated proteins were captured on glass filters and counted. These investigations revealed that lamotrigine had no impact on [$^{35}$S]-methionine incorporation, even when cells were treated with 8× MIC of lamotrigine (*Figure 6A, C*, black dots). Similarly, cells treated with these same concentrations of lamotrigine at 37°C for three doublings did not display any inhibition of translation (*Figure 6—figure supplement 1A*). We found that when cells were treated with 8× MIC of tetracycline, chloramphenicol, and erythromycin, there was a marked decrease in protein labeling. As expected given its known mechanism of action, cells treated with 8× MIC of vancomycin did not display inhibition of protein biosynthesis after 2.6 hr of treatment.

To determine if lamotrigine had a direct effect on protein biosynthesis in vitro, we employed a commercially available *E. coli* K-12 cell-free transcription/translation system producing luciferase. Reactions in the presence of 8× MIC of lamotrigine and the aforementioned antibiotics were incubated at 15°C for 4 hr (see *Figure 6—figure supplement 1C* for in vitro translation kinetics at 15°C), at which time luciferin was added to quantify the luciferase produced. As expected, all translation inhibitors blocked the production of luciferase while vancomycin did not (*Figure 6B*). Lamotrigine failed to block the production of luciferase at either 15°C or 37°C (*Figure 6B, C*, white dots, *Figure 6—figure supplement 1B*). To ensure that the in vitro translation assay required IF2 activity, we tested the effect of evernimicin, an oligosaccharide antibiotic known to inhibit IF2-dependent 70S initiation complex formation (*McNicholas et al., 2000*; *Belova et al., 2001*). Evernimicin prevented luciferase synthesis in vitro at both 15°C and 37°C (*Figure 6D*).

Having ruled out a direct effect on bulk protein biosynthesis, we wondered if lamotrigine might have a specific effect on r-protein synthesis that could lead to the accumulation of immature ribosome subunits. To test this, we measured the synthesis rate of each r-protein in vivo using a mass spectrometry-based pulse labeling technique. At 15°C, cells were grown in $^{14}$N-labeled M9 minimal media to mid-log phase at which point they were diluted twofold into $^{15}$N-labeled M9 media and concurrently treated with DMSO, lamotrigine, or chloramphenicol. Cells were harvested after 1 hr, 2.6 hr, 4 hr, 8 hr, and 16 hr and spiked with equal quantities of $^{15}$N-labeled 70S ribosomes as an internal reference standard. After cell lysis, these spiked samples were digested with trypsin for analysis by mass spectrometry. Using a Fourier transform deconvolution algorithm (*Sperling et al., 2008*; *Chen et al., 2012*), we independently quantified the r-proteins produced before the pulse ($^{14}$N) and those synthesized post-pulse (50% $^{15}$N) from the cellular lysate (*Figure 6—figure supplement 1D*).

Inspection of $^{14}$N abundance as a function of time revealed that most ribosomal proteins were stable over this time course (*Figure 6—figure supplement 1E,G,I*), consistent with our prior work (*Chen et al., 2012*). We then carefully inspected the rate of 50% $^{15}$N incorporation into each ribosomal protein in each treatment condition. Using a linear approximation of the synthesis rate based on

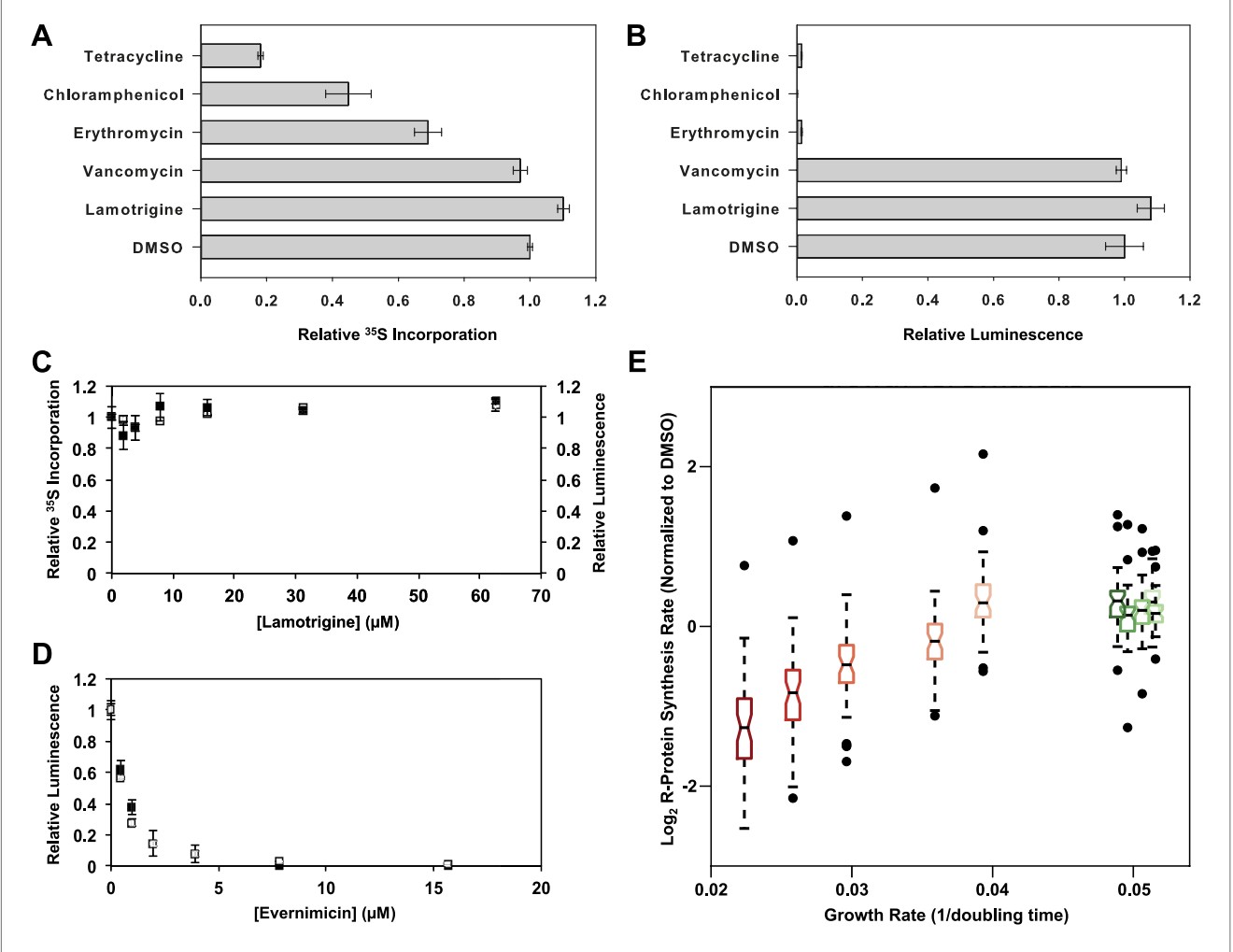

**Figure 6**. Accumulation of immature ribosomal subunits is not the result of translation inhibition. (**A**) [$^{35}$S]-methionine incorporation into early-log cells grown for 2.6 hr in M9 media at 15°C. Immediately prior to the radioactivity pulse, cultures were treated with 8× MIC of each antibiotic. [$^{35}$S]-methionine incorporation was quantified by liquid scintillation counting. Error bars represent the error of two biological replicates. (**B**) Cell-free coupled transcription/translation reactions in the presence of 8× MIC of each antibiotic. Samples were incubated at 15°C for 4 hr, at which time reactions were halted on ice, excess luciferin was added, and luminescence was monitored. Error bars represent the error of two biological replicates. (**C**) [$^{35}$S]-methionine incorporation (black dots) and cell-free luminescence (white dots) as a function of lamotrigine concentration. Samples were prepared as described in (**A**) and (**B**). (**D**) Cell-free coupled transcription/translation reactions in the presence of increasing concentrations of evernimicin at 37°C (black dots) and 15°C (white dots). Reactions were assembled and analyzed as in (**B**). (**E**) Analysis of growth rate and r-protein synthesis rate as a function of lamotrigine and chloramphenicol concentrations. Synthesis rates were determined for each ribosomal protein using quantitative mass spectrometry. For each condition, r-protein synthesis rates (50 measurements per treatment) are presented as a notched box and whisker plot centered at the growth rate (1/hr) observed for that treatment ('Materials and methods'). Black dots represent synthesis rates of individual proteins in excess of (1.5 × inner quartile range) of that data set. Light to dark shades of red represent 2×, 3×, 4×, 5×, and 6× MIC of chloramphenicol. Light to dark shades of green represent 2×, 3×, 4×, 5×, and 6× MIC of lamotrigine. Values were normalized to the DMSO control and log transformed.

The following source data and figure supplement are available for figure 6:

**Source data 1**. In vivo r-protein synthesis rates measured using qMS.

**Figure supplement 1**. Effects of lamotrigine on translation in *E. coli*.

the 4-, 8-, and 16-hr time points, we found that each protein was synthesized at a similar rate in the DMSO- and lamotrigine-treated cells up to 6× MIC. However, we found significant inhibition of r-protein synthesis with increasing concentrations of chloramphenicol, our positive control compound (***Figure 6E***, ***Figure 6—figure supplement 1F,H,J***, ***Figure 6—source data 1***).

## Immature ribosomal particles sediment as mature subunits upon removal of lamotrigine stress

To establish if the pre-30S and pre-50S particles that accumulate upon lamotrigine treatment represented immature subunits on pathway to maturity, we endeavored to monitor the impact of relieving inhibition by lamotrigine. We hypothesized that cells relieved of lamotrigine stress would assemble immature 30S and 50S particles into mature 30S and 50S subunits. *E. coli* was grown to early-log phase in LB media at 15°C and treated with either 2× MIC of lamotrigine or DMSO as a mock treatment. After 5 min, [$^{14}$C]-uridine was added and the cells were grown an additional 3 hr, at which point cells were pelleted, washed, and resuspended in fresh LB media supplemented with a 1000-fold excess of non-labeled uridine. Cells were harvested immediately preceding the chase, and after 30 min, 1 hr, 2 hr, and 3 hr of this chase period (*Figure 7A*).

In each DMSO-treated sample (*Figure 7B*), we found significant quantities of [$^{14}$C]-uridine-labeled 30S and 50S subunits. Because the quantity of labeled subunits did not change as a function of the length of the chase, these particles likely represent fully mature subunits that have simply dissociated. Interestingly, DMSO-treated cells harvested immediately after the 3-hr pulse and before the addition of the chase show a slight decrease in the levels of complete 70S ribosomes relative to any of the samples harvested post-chase (*Figure 7B*). This small but significant change likely results from the presence of an intracellular pool of [$^{14}$C]-uridine, which is incorporated into 70S particles during the initial 30-min chase. This pool may consist of free nucleotides that are not washed away during the chase or, as described previously, may exist as transcribed rRNA that has not completed the assembly process (*Chen et al., 2012*; *Chen and Williamson, 2013*). Cells harvested at subsequent times during the chase period showed no change in the quantities of 30S, 50S, and 70S particles, indicating that all newly synthesized rRNA is incorporating exclusively non-labeled uridine. This result allowed us to analyze the maturation of the lamotrigine-induced pre-30S and pre-50S particles, confident that they were generated during the initial pulse and not synthesized de novo between 30 min and 3 hr post-pulse.

We next analyzed the ability of cells treated with lamotrigine to process pre-30S and pre-50S particles (*Figure 7C*). Cells harvested immediately after the 3-hr pulse period displayed a significant accumulation of pre-30S and pre-50S material. As shown earlier (*Figure 3F*), we also noticed a large decrease in the relative accumulation of 70S ribosomes during drug treatment. Some 30 min after removal of lamotrigine and non-labeled uridine chase, the relative proportions of ribosomal particles began to adjust. Specifically, the levels of pre-30S and pre-50S particles decreased with a corresponding increase in 70S ribosomes. This trend continued throughout the 3-hr chase period, after which there were no apparent differences between DMSO-treated and lamotrigine-treated cells. Interestingly, after 1 hr of non-labeled uridine chase, a cluster of three particles that sedimented at approximately 40S (discussed above as the pre-50S), 45S, and 50S appeared. With each successive time point, the levels of 50S increased at the expense of the other particles, suggesting our time course had captured cells actively assembling the pre-50S particles into 50S subunits.

## Discussion

Understanding bacterial ribosome assembly has proven to be a challenging undertaking. Involving nearly 60 protein factors, the process is rapid, highly efficient, and studies to date suggest that assembly intermediates are elusive and do not accumulate in significant amounts (*Mulder et al., 2010*; *Chen et al., 2012*). The genetic inactivation of ribosome biogenesis factors has provided an opportunity to perturb the process in order to better understand the action of chemical modification and chaperone functions in the assembly process (*Shajani et al., 2011*). Nevertheless, many of these factors are essential and resist genetic manipulation. Further, genetic inactivation has poor temporal resolution and is not ideally suited to probe the coordinated action of these factors in time and space. Indeed, our understanding of ribosome function has benefited enormously from a great number and variety of small molecule probes of chemical and conformational steps of protein translation (*Tenson and Mankin, 2006*; *Wilson, 2009*). Chemical inhibitors of the assembly process would similarly provide important new probes of ribosome biogenesis. Herein, we report the discovery and characterization of a small molecule inhibitor of ribosome assembly in *E. coli* under cold temperature growth conditions. The inhibitor, lamotrigine, is a widely available anticonvulsant drug whose target in *E. coli* is domain II of the initiation factor IF2. In all, this work provides the first small molecule probe of ribosome assembly and points to a novel role for IF2 in *E. coli* ribosome biogenesis.

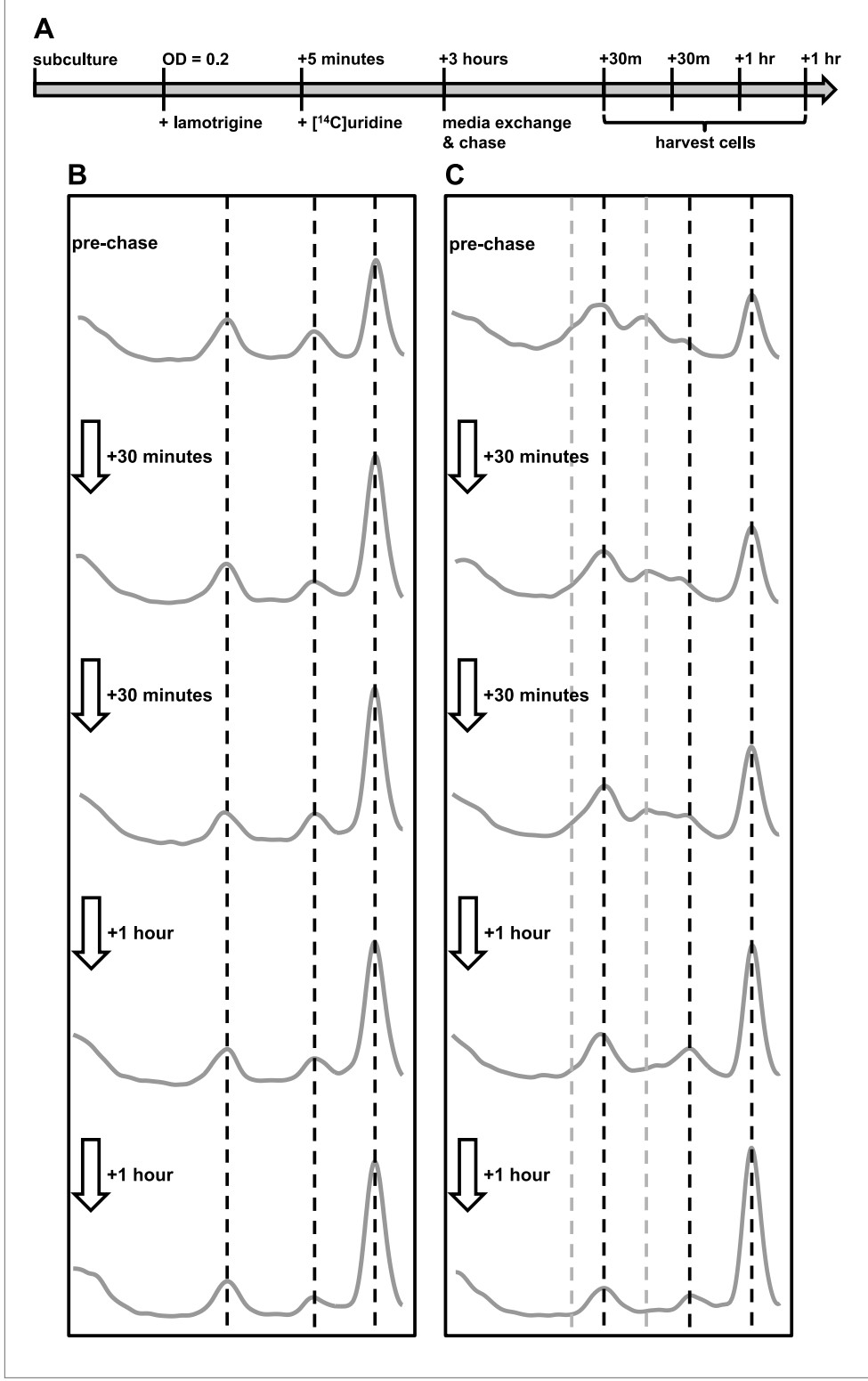

**Figure 7**. Immature ribosomal particles sediment as mature subunits upon removal of lamotrigine stress. Particles were analyzed by sedimentation over sucrose gradients and analyzed with radioactivity detection. (**A**) Experimental design of pulse-chase analysis of *E. coli* treated with 2× MIC lamotrigine. (**B**) Cells were treated with DMSO and concurrently pulsed with [¹⁴C]-uridine for 3 hr in LB media at 15°C prior to media exchange and unlabeled uridine chase. This time course reveals that no additional radiolabel was incorporated

*Figure 7. Continued on next page*

Figure 7. Continued

into ribosomal subunits during the chase period. (**C**) Cells were treated as in (**B**), except with 2× MIC lamotrigine in place of DMSO, revealing that radiolabeled pre-30S and pre-50S particles matured to 30S and 50S particles over the duration of chase period.

To find small molecule inhibitors of ribosome assembly, we developed a cell-based platform to first enrich for inhibitors of ribosome assembly and function by screening for compounds that led to a cold sensitive growth phenotype. The screen was inspired by numerous previous reports of cold sensitive mutants in ribosome-related genes and was validated with a screen of the *E. coli* Keio collection. In our screen of the Keio collection, ribosome genes were overwhelmingly enriched and, of the known cold sensitive ribosome biogenesis genes, we were successful in identifying the vast majority of these. We next screened a diverse collection of ~30,000 small molecules, including many known drugs and bioactive compounds, to identify growth inhibitory compounds with increased potency at 15°C relative to 37°C. Of 38 structurally diverse active compounds from this screen, lamotrigine induced the most profound cold sensitive growth inhibition. Sedimentation analysis revealed that lamotrigine induced the rapid accumulation of non-native ribosomal particles with apparent sedimentation rates of ~25S (pre-30S) and ~40S (pre-50S), prompting an in-depth analysis of their composition. Using 5′ primer extension of rRNA and r-protein mass spectrometry, these particles were found to be immature 30S and 50S subunits, respectively. These immature subunits lacked r-proteins associated with the neck of the 30S subunit and the body of the 50S subunit around the L1 arm, central protuberance, and L11 arm (***Figure 4—figure supplement 2C,D***). With these regions encompassing the functional centers of each subunit, it is tempting to speculate that lamotrigine may perturb late steps in 30S and 50S subunit assembly. It has recently been suggested that these sites are among the last to mature (***Jomaa et al., 2011***; ***Guo et al., 2013***; ***Li et al., 2013***; ***Jomaa et al., 2014***) consistent with this hypothesis.

Whole genome sequencing of spontaneous suppressor mutants capable of robust growth in the presence of lamotrigine revealed mutations in domain II near the N-terminus of initiation factor IF2. Further, in vitro binding studies indicated that lamotrigine binds to IF2 in a nucleotide-dependent fashion and that suppressor mutations abrogated binding. Interestingly, domain II is conserved solely among IF2 proteins from the Enterobacteriaceae and has yet to be assigned a definitive function. Indeed, only one study suggests a role for this region in binding strongly to 30S, 50S, and 70S ribosomal particles relative to the other Enterobacteriaceae IF2 domains (***Moreno et al., 1999***). While lamotrigine treatment resulted in the rapid accumulation of immature 30S and 50S subunits, the target IF2 led us to wonder if lamotrigine might be an inhibitor of protein translation. We speculated that the observed ribosome biogenesis phenotype might be an indirect effect of blocking r-protein production, as described previously for antibiotics known to inhibit translation (***Siibak et al., 2009***, ***2011***; ***Sykes et al., 2010***). Using multiple orthogonal approaches, both in vitro and in vivo, we were unable to detect any translational inhibition by lamotrigine even when using concentrations far above the MIC. These results were in contrast to assays with known translational inhibitors, including evernimicin, a known inhibitor of IF2-dependent 70S initiation complex formation.

The finding that IF2 is the target of lamotrigine is intriguing in light of emerging information on the role of its eukaryotic counterpart, eIF5B, in 40S subunit assembly in yeast (***Lebaron et al., 2012***; ***Strunk et al., 2012***). In *Saccharomyces cerevisiae*, eIF5B associates with immature 40S subunits in a translation-like checkpoint, wherein immature 40S particles bind to mature 60S subunits prior to final maturation. Given that pre-30S and pre-50S particles accumulate during lamotrigine stress, a bacterial model may include association of two immature particles prior to maturation of the functional centers within the 30S and 50S subunits. Alternatively, it is possible that IF2 is involved in the maturation of 30S and 50S subunits independently. Regardless of the precise events mediated by IF2, our data strongly support a central role for the enigmatic and divergent domain II of *E. coli* IF2 in the assembly of both subunits. Interestingly, our observations help rationalize the previously unexplained finding that overexpression of IF2 in a *ΔyjeQ* background of *E. coli* partially suppresses the mutant slow growth phenotype and restores ribosome profiles to wild type (***Campbell and Brown, 2008***). Similarly to lamotrigine-treated cells, the 30S particles of cells lacking YjeQ display significantly depleted occupancy of S21, S1, S2, and S3 (***Jomaa et al., 2011***), suggesting that IF2 may perform an overlapping role in late 30S maturation during cold stress. Our results also parallel work dating back almost two decades,

which showed that truncation of the N-terminus of *E. coli* IF2, containing domain II led to cold sensitive growth (*Laalami et al., 1991a*, *1991b*). With much of the machinery involved in ribosome biogenesis (*Bharat et al., 2006*; *Schaefer et al., 2006*) and translation (*Anger et al., 2013*) conserved among bacteria and eukaryotes, the maintenance of IF2/eIF5B function in ribosome biogenesis through evolution is surely plausible. Given that domain II of IF2 is highly divergent, this work raises questions of how diverse bacterial species carry out the temperature-dependent functions of IF2 described herein. Domain II may have a purpose that is uniquely important to the Enterobacteriaceae under cold stress or, alternatively, species-specific proteins that mimic the N-terminus of IF2 from Enterobacteriaceae may perform this activity.

Taken together, this work establishes lamotrigine as a first-in-class small molecule inhibitor of bacterial ribosome biogenesis. Moreover, we have identified domain II of IF2 as the molecular target of lamotrigine, suggesting an as-yet-uncharacterized ribosome assembly function for this canonical translation initiation factor. We posit that lamotrigine will serve as an important tool in expanding our understanding of the molecular details of IF2 in ribosome biogenesis and functions as a proof-of-concept molecule in the development of novel antibiotics.

## Materials and methods

### Screening for cold sensitivity

Overnight cultures of *E. coli* BW25113 (including Keio strains) grown in LB media at 37°C were diluted 1/1000 in fresh LB, and incubated at 15°C (48 hr) and 37°C (24 hr) in duplicate without shaking in a final volume of 100 µl. Cells were grown in Corning (Corning, NY) Costar 96-well clear-bottom plates. For the small molecule screen, compounds were added to *E. coli* BW25113 to a final concentration of 10 µM. All screens were performed in duplicate. Molecules, dissolved in DMSO, were sourced from ChemBridge (San Diego, CA), Maybridge (Waltham, MA), MicroSource Discovery Systems (Gaylordsville, CT), Prestwick Chemicals (Washington, DC), and Biomol-Enzo Life Sciences (Farmingdale, NY). Liquid handling was performed using a Beckman Coulter (Brea, CA) FX$^P$ Laboratory Automated Workstation. After incubation, plates were read using a Perkin Elmer (Waltham, MA) EnVision plate reader at 600 nm.

### Sucrose density gradient analysis

25 ml cultures of early-log *E. coli* BW25113 (OD = 0.2) grown in LB media at 15°C were treated with the appropriate concentration of each antibiotic (purchased from Sigma, St. Louis, MO) and, when applicable, pulse labeled with [$^{14}$C]-uridine (purchased from American Radiolabeled Chemicals, St. Louis, MO) to a final concentration of 0.2 µCi/ml (specific activity 55 mCi/mmol). Cells were incubated as necessary, harvested by centrifugation, and lysed using a Constant Systems (Daventry, England) cell disruptor at 13 kpsi in 3 ml ice-cold ribosome buffer (20 mM Tris–HCl, pH 7.0, 10.5 mM MgOAc, 100 mM NH$_4$Cl, 3 mM β-mercaptoethanol). Cell lysates were clarified using a Beckman Coulter MLA-80 rotor at 24,000 rpm for 45 min, at which time they were loaded onto 35 ml 10–40% sucrose gradients and centrifuged for 18 hr at 18,700 rpm in a Thermo (Waltham, MA) SureSpin rotor. The volume of lysate added to each gradient was adjusted based on OD$_{600}$ of the DMSO-treated control culture to ensure reproducibility across experiments. Gradients were either fractionated using an AKTA Prime FPLC (GE Healthcare, Little Chalfont, England) outfitted with a continuous flow UV cell at 260 nm or analyzed via continuous flow UV and scintillation counting using an AKTA Prime FPLC in series with a Perkin Elmer 150TR flow scintillation analyzer.

### 5′ primer extension analysis of rRNA

Sucrose density gradients, loaded with clarified cell lysates normalized for OD$_{600}$, were ran as described above. Total rRNA from 500 µl sucrose gradient fractions was purified using phenol chloroform extraction followed by sodium acetate precipitation and dissolved in 5 µl of water. 1 µl of rRNA from each sample was added to 9 µl of water and 1 µl (2.4 pmol) of the necessary primer was added. Each 11 µl reaction was incubated at 80°C for 10 min and allowed to cool to room temperature in order to denature the rRNA. rRNA was subsequently reverse transcribed at 45°C for 24 hr using RevertAid H Minus Reverse Transcriptase from Thermo Scientific in a reaction volume of 20 µl according to the manufacturers instructions. cDNA products from each reaction were precipitated using sodium acetate and 90% ethanol and washed once in 70% ethanol. Purified cDNA samples were analyzed via capillary electrophoresis using a GeneScan 350 TAMRA size standard

(Thermo Scientific). 16S rRNA and 23S rRNA primers containing a 5′ 6-oxyfluorescein marker were purchased from Sigma. 16S rRNA primer sequence: 5′-CTGTTACCGTTCGACTTG-3′. 23S rRNA primer sequence: 5′-CTTATCGCAGATTAGCACG-3′.

## R-protein quantitation using mass spectrometry

Reference standard ribosomal particles were prepared by growing *E. coli* strain NCM3722 in supplemented M9 (48 mM $Na_2HPO_4$, 22 mM $KH_2PO_4$, 8.5 mM NaCl, 10 mM $MgCl_2$, 10 mM $MgSO_4$, 5.6 mM glucose, 50 µM $Na_3$·EDTA, 25 mM $CaCl_2$, 50 µM $FeCl_3$, 0.5 µM $ZnSO_4$, 0.5 µM $CuSO_4$, 0.5 µM $MnSO_4$, 0.5 µM $CoCl_2$, 0.04 µM d-biotin, 0.02 µM folic acid, 0.08 µM vitamin B1, 0.11 µM calcium pantothenate, 0.4 nM vitamin B12, 0.2 µM nicotinamide, and 0.07 µM riboflavin) bearing 7.6 mM of either $^{14}N$ or $^{15}N$-labeled $(NH_4)_2SO_4$. Cells were harvested at OD = 0.5 and lysed in buffer A (20 mM Tris–HCl, 100 mM NH4Cl, 10 mM MgCl2, 0.5 mM EDTA, 6 mM β-mercaptoethanol; pH 7.5) using a mini bead beater. Clarified lysates (5 ml) were layered above a 5 ml sucrose cushion (20 mM Tris–HCl, 500 mM NH4Cl, 10 mM MgCl2, 0.5 mM EDTA, 6 mM β-mercaptoethanol, 37% sucrose; pH 7.5) and were spun for 22 hr at 37.2k rpm in a Ti 70.1 rotor. Pellets bearing 70S ribosomes were solubilized in buffer A at 4°C and saved at −80°C. Lamotrigine- and DMSO-treated ribosomal particles were separated on a sucrose density gradient and fractions were collected as described above. A mixed reference standard bearing 10 pmol of $^{14}N$-labeled and 30 pmol of $^{15}N$-labeled 70S ribosomal particles was added to 20 pmol of each experimental fraction. The use of this mixed reference ensured that every $^{15}N$-labeled peptide bore a $^{14}N$-labeled peptide pair irrespective of the abundance of that peptide in the experimental sample. Additionally, one sample bearing only the reference standard was mixed with an equal volume of buffer A. These samples were then prepared for LC/MS via precipitation, reduction, alkylation, and tryptic digestion as described previously (*Jomaa et al., 2014*). Peptides were eluted from a C18 column using a concave acetonitrile gradient and detected using first an Agilent (Santa Clara, CA) G1969A ESI-TOF and second, to improve proteomic coverage and to identify non-ribosomal proteins, using an AB/Sciex (Framingham, MA) 5600 Triple-TOF run in $MS^2$ mode. In each case, the entire isotope distribution of each extracted $MS^1$ spectrum was fit using a Least Squares Fourier Transform Convolution algorithm (*Sperling et al., 2008*) providing accurate quantitation of the $^{14}N$ and $^{15}N$ species' abundance. To account for the reference standard's contribution to the measured $^{14}N$ peptide abundance, each spectrum was normalized using the paired $^{15}N$ abundance. Having measured the reference standard alone in triplicate, we then subtracted these normalized spectra, resulting in the corrected peptide abundance for each peptide in each experimental sample. Data sets from the ESI-TOF and Triple-TOF were merged and filtered for interference from co-eluting peptides. As a proof of principle, a series of standards bearing various quantities of $^{14}N$-labeled 70S particles were also analyzed to assess the linearity of our detection technique (*Gulati et al., 2014*). Non-ribosomal proteins were quantified across the gradient using the aforementioned $MS^2$ Triple-TOF data sets. These data sets were acquired as IDA experiments with 200 ms $MS^1$ scans followed by 50 $MS^2$ scans, each with 50 ms of ion accumulation. Precursor ions were excluded from $MS^2$ analysis 12 s after one occurrence. In each fraction, spectral counts for each non-ribosomal protein were normalized to the total number of spectral counts in that fraction. These values were then normalized to the maximal spectral counts in any gradient fraction, and the occupancy profile was smoothed using a 3-fraction sliding Gaussian window.

## $^{15}N$ pulse labeling

30 ml mid-log cultures of *E. coli* BW25113 (OD = 0.4) grown in M9 media were pulsed with 50% $^{15}N$ by adding 30 ml of M9 containing $^{15}N$ ammonium chloride as the sole nitrogen source. Cells were introduced to the necessary concentrations of chloramphenicol or lamotrigine during the pulse by supplementing the $^{15}N$ M9 with antibiotic. Cells were pulsed for 0 hr, 1 hr, 2.6 hr, 4 hr, 8 hr, and 16 hr, at which times 10 ml of each culture was removed and the cells harvested via centrifugation. Cell pellets were frozen at −80°C prior to processing for mass spectrometry.

## R-protein synthesis measured by pulse-labeling quantitative mass spectrometry

Pulse-labeled cells were spiked with 20 pmol of $^{15}N$-labeled 70S ribosomal particles and prepared for analysis on the ESI-TOF as described above. Extracted $MS^1$ spectra were fit using Least Squares Fourier Transform Convolution algorithm with three species: 0% $^{15}N$ (pre-pulse), 50% $^{15}N$ (post-pulse), and 100% $^{15}N$ (*Sperling et al., 2008*). Each species was normalized to the reference resulting in the following: pre-pulse material [0%/100%], post-pulse synthesis [50%/100%], and total material

([0% + 50%]/100%). Synthesis rates were calculated for each r-protein independently by fitting the median post-pulse synthesis measurement for the 0, 4, 8, 16-hr time points to a line. The synthesis rate of each of r-protein in the lamotrigine or chloramphenicol treatment was normalized to that of the DMSO treatment and log-transformed. The resultant values were presented as notched box and whisker plots, centered at the growth rate for each treatment (*Figure 6E*). Whiskers extend to the most extreme data point within 1.5 times the inner quartile range (IQR) whereas notches extend from the median $1.57 \times IQR/(number of points)^{1/2}$. All aforementioned data analysis was performed using a series of Python scripts available at https://github.com/joeydavis/StokesDavis_eLife_2014.

### Lamotrigine suppressor isolation and whole genome sequencing

Dense overnight cultures of *E. coli* BW25113 were diluted 1/1000 in 10 ml of LB media supplemented with 39 µM lamotrigine and grown at 15°C until cultures became dense. This occurred after 7 days of incubation. Potential suppressor clones from these cultures were subsequently passaged three times on LB agar. Single colonies from LB agar plates were then re-streaked onto LB agar supplemented with 39 µM lamotrigine to assess mutation stability and purify individual suppressor clones. Individual colonies were isolated based on colony diameter and analyzed via UV absorbance at 600 nm in liquid LB media to determine growth kinetics and lamotrigine MICs at 15°C. Kinetic growth assays were conducted in a temperature-controlled Tecan (Mannedorf, Switzerland) Sunrise plate reader. The ribosomal particles of these clones were analyzed using sucrose density gradient centrifugation as described above. Genomic DNA from wild-type *E. coli* BW25113 and lamotrigine suppressor mutants were purified using a Qiagen (Venlo, Netherlands) Gentra Puregene kit and sequenced using an Illumina (San Diego, CA) MiSeq platform. Paired-end 250 bp read data for wild type and mutant samples were aligned to the *E. coli* MG1655 chromosome (NC_000913) using BowTie2, and mutations were visualized and annotated using BreSeq and Tablet.

### [³H]-lamotrigine binding assays

Wild type and mutant #3 *E. coli infB* genes were cloned into the pDEST17 plasmid containing an N-terminal His tag using the Invitrogen (Carlsbad, CA) Gateway cloning system. Protein expression was conducted in *E. coli* BL21-AI cells grown in LB at 15°C. Expression was induced at OD ~0.6 using 0.2% arabinose, and cells were harvested after 16 hr of induction. Cells were lysed using a Constant Systems cell disruptor at 20 kpsi in IF2 lysis buffer (50 mM HEPES–KOH, pH 7.4, 1 M $NH_4Cl$, 10 mM $MgCl_2$, 0.1% Triton X-100, 7 mM β-mercaptoethanol) containing EDTA-free protease inhibitor tablets from Roche (Basel, Switzerland). Cell lysates were clarified via centrifugation at 20,000 rpm for 45 min in a Beckman Coulter JA-25.50 rotor. Clarified lysates were loaded onto a 1 ml GE Healthcare HisTrap FF column and eluted with IF2 elution buffer (50 mM HEPES–KOH, pH 7.4, 1 M $NH_4Cl$, 10 mM $MgCl_2$, 7 mM β-mercaptoethanol, 400 mM imidazole). Purified wild type and mutant IF2 was buffer exchanged into ice-cold ribosome buffer using an Amicon (Millipore, Billerica, MA) Ultracel filtration unit with a 50 kDa cutoff filter. 50 µl reactions containing 2 mg/ml BSA, 20 µM IF2, 200 nM [³H]-lamotrigine (specific activity 5 Ci/mmol; purchased from American Radiolabeled Chemicals), and 30 mM G-nucleotide (purchased from Sigma) were incubated at 15°C in ribosome buffer for 3 hr. Reactions were loaded onto 200 µl pre-wet Sephadex G-25 resin beds (resin purchased from GE Healthcare) and centrifuged at $400 \times g$ for 3 min. Flow-through samples were scintillation counted using Perkin Elmer Ultima Gold scintillation fluid.

### In vivo translation analysis

1 ml early-log cultures of *E. coli* BW25113 (OD = 0.2) grown in M9 media were treated with 8× MIC of various antibiotics and were concurrently pulse labeled with [³⁵S]-methionine (purchased from Perkin Elmer) to a final concentration of 5 µCi/ml (specific activity 1175 Ci/mmol). Cells were incubated for 2.6 hr at 15°C, at which time they were harvested via centrifugation and washed twice in 1 ml 0.85% saline. Cells were lysed using 100 µl Millipore BugBuster Master Mix reagent and proteins were precipitated using 25 µl ice-cold 25% TCA. Protein pellets were then washed twice in 25 µl ice-cold 10% TCA and passed through Whatman (GE Healthcare) GF/C filters using a Millipore vacuum manifold. Filters were washed three times in ice-cold 10% TCA, dried overnight at room temperature, and scintillation counted using Perkin Elmer Ultima Gold scintillation fluid.

### In vitro translation analysis

Cell-free translation was conducted using the *E. coli* S30 transcription–translation system for circular DNA from Promega (Finchburg, WI) according the manufacturer's instructions. 10 µl reactions containing

the necessary antibiotics and plasmid DNA encoding the firefly luciferase gene were incubated either at 15°C for 4 hr or 37°C for 1 hr. Reactions were halted on ice for 5 min prior to addition of 25 μl of room-temperature luciferin (purchased from Promega). Immediately after the addition of luciferin, samples were analyzed for luminescence output in a Nunc (Roskilde, Denmark) 384-well clear bottom plate using a Tecan Ultra Evolution luminometer.

## Accession numbers

The GenBank accession numbers for the IF2 variants described in this paper are KJ752767 (wild-type *E. coli* BW25113 IF2); KJ52768 (mutant #1 IF2); KJ752769 (mutant #2 IF2); KJ752770 (mutant #3 IF2); and KJ752771 (mutant #4 IF2).

## Acknowledgements

We thank Dr Mike Surette and Nicholas Waglechner from McMaster University for assistance with genome analysis; members of the Center for Microbial Chemical Biology at McMaster University for technical assistance with screening robotics; Dr Stephan Nord, formerly in the lab of EDB, for assistance in 5′ primer extension and lamotrigine suppressor generation; and Kella Kapnisi, formerly in the lab of EDB, for assistance with small molecule screening.

## Additional information

### Funding

| Funder | Grant reference number | Author |
| --- | --- | --- |
| Natural Sciences and Engineering Research Council of Canada | | Eric D Brown |
| Canada Research Chairs | | Eric D Brown |
| National Institutes of Health | R37-GM053757 | James R Williamson |
| Canadian Institutes of Health Research | | Jonathan M Stokes |
| Ontario Council on Graduate Studies, Council of Ontario Universities | | Jonathan M Stokes |
| Jane Coffin Childs Memorial Fund for Medical Research | | Joseph H Davis |

The funders had no role in study design, data collection and interpretation, or the decision to submit the work for publication.

### Author contributions

JMS, JHD, Conception and design, Acquisition of data, Analysis and interpretation of data, Drafting or revising the article; CSM, Conception and design, Acquisition of data, Analysis and interpretation of data; JRW, Analysis and interpretation of data, Drafting or revising the article; EDB, Conception and design, Analysis and interpretation of data, Drafting or revising the article

## Additional files

### Supplementary files

• Supplementary file 1. Related to *Figure 1*. Identification of previously reported cold sensitive ribosome biogenesis genes. All genes presented have been reported as cold sensitive. The Comments column describes whether each gene deletion strain displayed cold sensitivity in our screen of the Keio collection.

• Supplementary file 2. Related to *Figure 5*. Temperature-dependent activity of lamotrigine across bacterial species.

## Major datasets

The following dataset was generated:

| Author(s) | Year | Dataset title | Dataset ID and/or URL | Database, license, and accessibility information |
|---|---|---|---|---|
| Stokes JM, Davis JH, Mangat CS, Williamson JR, Brown ED | 2014 | Discovery of a small molecule that inhibits bacterial ribosome biogenesis | KJ752767; KJ52768; KJ752769; KJ752770; KJ752771 | Publicly available at NCBI GenBank (http://www.ncbi.nlm.nih.gov/genbank). |

The following previously published datasets were used:

| Author(s) | Year | Dataset title | Dataset ID and/or URL | Database, license, and accessibility information |
|---|---|---|---|---|
| | | Escherichia coli str. K-12 substr. MG1655 chromosome, complete genome | http://www.ncbi.nlm.nih.gov/nuccore/NC_000913 | Publicly available at NCBI GenBank. |
| Schuwirth BS, Borovinskaya MA, Hau CW, Zhang W, Vila-Sanjurjo A, Holton JM, Cate JH | 2005 | Structures of the bacterial ribosome at 3.5Å resolution | http://www.pdb.org/pdb/explore/explore.do?structureId=2AVY | Publicly available at RCSB Protein Data Bank. |
| Schmeing TM, Voorhees RM, Kelley AC, Ramakrishnan V | 2011 | How mutations in tRNA distant from the anticodon affect the fidelity of decoding | http://www.pdb.org/pdb/explore/explore.do?structureId=2Y11 | Publicly available at RCSB Protein Data Bank. |
| Laursen BS, Mortensen KK, Sperling-Petersen HU, Hoffman DW | 2003 | A conserved structural motif at the N terminus of bacterial translation initiation factor IF2 | http://www.pdb.org/pdb/explore/explore.do?structureId=1ND9 | Publicly available at RCSB Protein Data Bank. |
| Roll-Mecak A, Cao C, Dever TE, Burley SK | 2000 | X-Ray structures of the universal translation initiation factor IF2/eIF5B: conformational changes on GDP and GTP binding | http://www.pdb.org/pdb/explore/explore.do?structureId=1G7R | Publicly available at RCSB Protein Data Bank. |

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
