## [Decision Letter]

Thank you for sending your work entitled “Discovery of a small molecule that inhibits
bacterial ribosome biogenesis” for consideration at *eLife.* Your article
has been favorably evaluated by Michael Marletta (Senior editor), Roberto Kolter
(Reviewing editor), and 2 reviewers.

The Reviewing editor and the other reviewers discussed their comments before we reached
this decision, and the Reviewing editor has assembled the following comments to help you
prepare a revised submission.

In this manuscript Stoke et al use a chemogenetic screen to discover a small molecule,
lamotrigine (lam) that induces cold-sensitivity in *E. coli*. The authors
then proceed to demonstrate that lam perturbs assembly of 30S and 50S ribosomes.
Suppressor analysis and biochemical analysis demonstrate that the translation initiation
factor IF2 is the target of lam. Surprisingly though, lam does not perturb
translation.

This is an interesting and exciting paper with intriguing implications. However, it was
felt that some minor additional experimental support would solidify the claims. In
particular, growth curves and/or serial dilution plating of bacterial strains at the
permissive (37°C), nonpermissive (15°C), and various temperatures in between using media
(LB and M9) containing various concentrations of lamotrigene (including concentrations
used in various experiments) should be presented. 15°C is very low temperature for
growth of *E. coli*. The doubling time for the DMSO/wild-type control at
15°C is about 12x that at 37°C and this raises the question of how much cold shock plays
into these data. It is unclear why such a low temperature was chosen for this study.
Results from the proposed simple experiments would allow more confident correlation of
defects in ribosome biogenesis with effects on cell growth.

In addition, there are several places where the presentation of the data is difficult to
follow and thus the authors should modify the text in response to the minor comments
listed below.

1) Presumably the cold sensitivity factor is the ratio of growth at 37°C, relative to
15°C, relative to wild type?

2) Error bars would be desirable in Figure 1
considering that the % cold sensitivity for the Co-enzyme metabolism gene class was only
∼2 fold less than the Translation, ribosome structure, and biogenesis gene class. This
difference was the foundation for all the subsequent work.

3) Figure 4 is extremely difficult to follow:
What is the difference in the experiment between 4B and C? It seems that panel B says
that there is no measurable accumulation of pre-16S rRNA, but pre-23S rRNA is
accumulated in the pre-30S and the pre-50S peak. Nevertheless, panel C says that pre-16S
rRNA also accumulates. Where are these discrepancies coming from? Also, perhaps clearer
labeling could help. I would call p16S, pre-16S; perhaps one could call +7 and +3,
pre-pre50S, and pre-50S, respectively; or perhaps pre-50S+3 or pre-50S+7; it is also not
explained in the legend where the +7 is coming from; it took me a while to figure that
out.

4) What is Figure 4 supposed to show; what is on
the x-axis? Neither the text nor the legend is helpful.

5) It is hard to tell if S3 is depleted from the pre-30S as claimed. I'd say no...

6) The effects on 50S maturation seem to be far more drastic than the effects on 30S
maturation; perhaps that should be discussed.

7) It would be nice to show a structure of mature 30S and 50S illustrating which of the
RPs are missing. Also, the authors point out that IF2 overexpression rescues the YjeQ
deletion; interestingly the YjeQ deletion has a very similar phenotype in terms of
depleted proteins.

8) The data indicates a maturation-specific function of the drug, and perhaps even of
domain II. This should be discussed, and a domainII-specific function should be tested
by complementation with IF2 from divergent species at low temperature.

---

## [Author Response]

*Growth curves and/or serial dilution plating of bacterial strains at the
permissive (37°C), nonpermissive (15°C), and various temperatures in between using
media (LB and M9) containing various concentrations of lamotrigene (including
concentrations used in various experiments) should be presented. 15°C is very low
temperature for growth of E. coli. The doubling time for the DMSO/wild-type control
at 15°C is about 12x that at 37*^*°*^*C and
this raises the question of how much cold shock plays into these data. It is unclear
why such a low temperature was chosen for this study. Results from the proposed
simple experiments would allow more confident correlation of defects in ribosome
biogenesis with effects on cell growth*.

These experiments have been completed, and the results can be found in Figure 2—figure supplement 2. We observed a
media-independent phenotype where lamotrigine activity was inversely proportional to
growth temperature. These results are consistent with the conclusion that lamotrigine
inhibits a cold sensitive event during ribosome biogenesis. As temperature decreases,
the apparent role of IF2 in ribosome biogenesis becomes increasingly essential, thus
increasing the growth inhibitory effect of lamotrigine. 15°C was selected in this study
to maximize cold-induced ribosome biogenesis phenotypes while still permitting cell
growth amenable to high-throughput screening in micro-well format. Based on the new data
shown in Figure 2—figure supplement 2, we
believe that lamotrigine would produce identical ribosome biogenesis phenotypes at
higher temperatures, albeit at correspondingly higher concentrations.

*1) Presumably the cold sensitivity factor is the ratio of growth at
37°C*, *relative to 15°C, relative to wild type?*

The cold sensitivity factor is defined in Figure legend 1 as the ratio of growth at 37°C
to growth at 15°C, normalized to the mean of all cold sensitivity factors calculated. In
response to the reviewers comment, we have also noted this normalization in the Results
section.

*2) Error bars would be desirable in*
Figure 1
*considering that the % cold sensitivity for the Co-enzyme metabolism gene class
was only ∼2 fold less than the Translation, ribosome structure, and biogenesis gene
class. This difference was the foundation for all the subsequent work*.

Figure 1 is a histogram showing the ratio of
cold sensitive gene deletion strains in each COG functional class to the total number of
non-essential *E. coli* genes in that same functional class. As such,
calculating error to determine statistical significance of these results is not
possible. Alternatively, we performed a permutation test to determine the probability of
observing a 21% hit rate in the translation, ribosome structure & biogenesis
functional class by chance. By permuting the classification assignments, we determined
that the 21% cold sensitive genes in the Translation class was significant with a
bootstrapped p-value < 1e^-6^. This p-value has been added to the legend of
Figure 1.

*3)*
Figure 4
*is extremely difficult to follow: What is the difference in the experiment
between 4B and C? It seems that panel B says that there is no measurable accumulation
of pre-16S rRNA, but pre-23S rRNA is accumulated in the pre-30S and the pre-50S peak.
Nevertheless, panel C says that pre-16S rRNA also accumulates. Where are these
discrepancies coming from? Also, perhaps clearer labeling could help. I would call
p16S, pre-16S; perhaps one could call +7 and +3, pre-pre50S, and pre-50S,
respectively; or perhaps pre-50S+3 or pre-50S+7; it is also not explained in the
legend where the +7 is coming from; it took me a while to figure that
out*.

This is indeed a complex figure and these comments have helped us to improve the clarity
of these panels. We have modified Figure 4, the
legend for Figure 4 and have re-written the
Results section describing the relationship between Figure 4. Briefly, Figure 4
illustrates the proportion of processed rRNA in various positions through the gradient.
This proportion of processed rRNA is defined as [immature rRNA/total rRNA] for each
species analyzed (either 16S or 23S). This is ideal for understanding the rRNA
processing efficiency of individual ribosomal particles. However, as was seen in the
pre-30S region of the gradient in lamotrigine-treated cells, we were able to purify
measurable quantities of both 16S and 23S species that were unprocessed. Thus, to
discern whether the dominant species residing within the pre-30S region was 30S or 50S
in nature, we needed to determine absolute quantities of 16S rRNA species and 23S rRNA
species (Figure 4) originating from this region.
Performing this experiment with 5’ primer extension of both 16S and 23S rRNA in parallel
from the same sucrose gradient fractions allowed us to conclude that the major species
in the pre-30S peak was an immature 30S particle.

*4) What is*
Figure 4
*is supposed to show; what is on the x-axis? Neither the text nor the legend is
helpful*.

We have greatly expanded the figure legend and our discussion of this figure. We have
also modified the legend for Figure 4—figure supplement 2 to avoid any additional confusion caused by figures of this form.
This panel is intended to show example r-protein occupancy data of small and large
subunit proteins (S3, S15, L28, L24) along sucrose gradients of DMSO- and
lamotrigine-treated cells. Blue represents early fractions in the gradient and red
indicates late fractions. The vertical axis quantifies r-protein occupancy. For each
protein, the horizontal axis spans different fractions along the sucrose gradient from
lowest density (blue; left) to highest density (red; right).

5) It is hard to tell if S3 is depleted from the pre-30S as claimed. I'd say
no...

Although the depletion effect on S3 is less pronounced than that on S2 or S21, we
contend that the protein is in fact depleted from the pre-30S peak in the lamotrigine
treated cells and the 30S peak in the DMSO-treated cells. The effect is most apparent by
comparing the intensity in fractions 1-4 of S3 with those from S8, S15, S6, and S18 in
Figure 4. Clearly the S3 abundance is lower
than that of S8, etc. Additionally, the decreased abundance of S3 can be seen in the
DMSO-treated cells by comparing the 30S peak intensities (fractions 7-9) of S3 to those
of S20, S11, S12, etc. In response to the reviewers concerns, we have modified the
Results section to emphasize the effect on S2 and S21 and to deemphasize the effect on
S3. Critically, this change has no effect on the overall conclusions of the
manuscript.

*6) The effects on 50S maturation seem to be far more drastic than the effects on
30S maturation; perhaps that should be discussed*.

We agree that the data demonstrate very strong effects of lamotrigine treatment on 50S
assembly. However, as evidenced by the significant accumulation of immature 16S rRNA in
the pre-30S fractions, the effects on 30S assembly are profound in their own right. In
the absence of a detailed mechanism for IF2 function in ribosome biogenesis, we have
chosen to give equal weight to the effects observed on the 30S and 50S subunits. We
contend that apparent assembly defects may be more pronounced in the large subunit
simply as a result of the size and complexity of this subunit relative to the 30S. As a
result of this complexity, even mild assembly inhibition may lead to the accumulation of
the discrete particle we observe. Indeed, such effects have been observed upon depletion
of a variety of assembly factors, some of which are non-essential. For example, deletion
of SrmB results in the accumulation of a discrete 40S particle despite relatively mild
effects on cellular growth rate (35).

*7) It would be nice to show a structure of mature 30S and 50S illustrating which
of the RPs are missing. Also, the authors point out that IF2 overexpression rescues
the YjeQ deletion; interestingly the YjeQ deletion has a very similar phenotype in
terms of depleted proteins*.

Images highlighting the depleted ribosomal proteins can be found in Figure 4—figure supplement 2 panels C, D. We have modified the
figure legends for these panels to clarify their content. We have also added a
discussion point hypothesizing the functional relationship between YjeQ and IF2 during
cold stress in *E. coli*.

*8) The data indicates a maturation-specific function of the drug, and perhaps
even of domain II. This should be discussed, and a domainII-specific function should
be tested by complementation with IF2 from divergent species at low
temperature*.

We postulated the potential role(s) of IF2 in ribosome biogenesis in the Discussion.
However, we attempted to keep this section concise in order to avoid over-interpretation
of our data. Further investigation of the role of domain II in ribosome biogenesis
through complementation is a great idea, and has previously been indirectly conducted.
Two thorough investigations by Laalami et al. (32; 51)
showed that truncations of the N-terminus of IF2 resulted in *E. coli*
strains that were incapable of growing at temperatures below 40°C. These results are in
agreement with the cold sensitive phenotype induced by lamotrigine, and strongly suggest
that complementing *E. coli* IF2 with the *infB* gene
product from divergent species lacking domain II would result in cells that similarly
display cold sensitivity.